# In-Silico Analysis Highlights the Existence in Members of *Burkholderia cepacia* Complex of a New Class of Adhesins Possessing Collagen-like Domains

**DOI:** 10.3390/microorganisms11051118

**Published:** 2023-04-25

**Authors:** Ricardo Estevens, Dalila Mil-Homens, Arsenio M. Fialho

**Affiliations:** 1Department of Bioengineering, Instituto Superior Técnico, University of Lisbon, Av. Rovisco Pais, 1049-001 Lisbon, Portugal; ricardo.estevens@hotmail.com (R.E.); dalilamil-homens@tecnico.ulisboa.pt (D.M.-H.); 2Institute for Bioengineering and Biosciences (iBB), Instituto Superior Técnico, University of Lisbon, Av. Rovisco Pais, 1049-001 Lisbon, Portugal; 3Institute for Health and Bioeconomic (i4HB), Instituto Superior Técnico, University of Lisbon, Av. Rovisco Pais, 1049-001 Lisbon, Portugal

**Keywords:** *B. cenocepacia*, cell adhesion, bacterial collagen-like middle domain, tandem repeats, intrinsically disordered proteins

## Abstract

*Burkholderia cenocepacia* is a multi-drug-resistant lung pathogen. This species synthesizes various virulence factors, among which cell-surface components (adhesins) are critical for establishing the contact with host cells. This work in the first part focuses on the current knowledge about the adhesion molecules described in this species. In the second part, through in silico approaches, we perform a comprehensive analysis of a group of unique bacterial proteins possessing collagen-like domains (CLDs) that are strikingly overrepresented in the *Burkholderia* species, representing a new putative class of adhesins. We identified 75 CLD-containing proteins in *Burkholderia cepacia* complex (Bcc) members (Bcc-CLPs). The phylogenetic analysis of Bcc-CLPs revealed the evolution of the core domain denominated “Bacterial collagen-like, middle region”. Our analysis remarkably shows that these proteins are formed by extensive sets of compositionally biased residues located within intrinsically disordered regions (IDR). Here, we discuss how IDR functions may increase their efficiency as adhesion factors. Finally, we provided an analysis of a set of five homologs identified in *B. cenocepacia* J2315. Thus, we propose the existence in Bcc of a new type of adhesion factors distinct from the described collagen-like proteins (CLPs) found in Gram-positive bacteria.

## 1. Introduction

The human respiratory pathogen *Burkholderia cenocepacia*: a model organism to study cellular adhesion.

*Burkholderia cenocepacia* is one out of at least twenty species included in the denominated *Burkholderia cepacia* complex (Bcc). This group of Gram-negative bacteria include closely related bacteria spread across multi environments due to their great adaptability [1]. They are isolated from soil or aquatic habitats, roots, and the leaves of plants. In such environments, Bcc members establish commensal associations with local microbiota. Moreover, some of the members show unique metabolic abilities namely as degraders of chemical pollutants and as atmospheric nitrogen fixers [2,3]. Besides that, members of the Bcc have emerged as problematic opportunistic pathogens in immunocompromised and cystic fibrosis (CF) patients [4,5]. *B. cenocepacia* and *Burkholderia multivorans* have been identified as the Bcc members more prevalent in CF infections [6]. They appear to be highly adapted to the respiratory tract through genomic plasticity and are prone to developing antibiotic resistance. CF patients are especially vulnerable to chronic lung infections caused by these pathogens, developing, in some cases, necrotizing pneumonia and septicemia, resulting in death [5]. Anti-burkholderial first line therapeutic options are limited to a few agents, and new drugs and alternative therapies are urgently needed. One potential way forward is using anti-adhesion therapies.

The phenomena of bacterial adhesion represent one of the clues for their colonization success. Based on unusually large genomes (3–4 replicons), Bcc members may have an extended repertoire of adhesion factors that are made available when they establish contact with abiotic or biotic surfaces [7]. However, the adhesion events in Bcc, specifically for the most pathogenic species, are poorly understood and many genes remain to be identified. It has become clear that the expression of adhesion factors has to be limited to time and environment and is tightly regulated, complex, and difficult to simulate under laboratory conditions. This observation reveals that the entire adhesiome of this pathogen remains far from being fully understood. 

To increase the knowledge about the existence of new *B. cenocepacia* adhesion factors, combinatorial computational and experimental procedures are suitable to be used. In silico analysis of genomic and transcriptomic data allows the screening and identification of candidate genes, through the prediction of the structure and function of their products. While the genomic approach relies merely on the information given by annotation, the use of transcriptomic data (RNA-Seq technology) uncovers the overexpressed genes associated with a specific physiological adhesion condition. These approaches have been used successfully for the identification of adhesion factors in various pathogens, such as *Streptococcus pyogenes* [8], *Klebsiella pneumoniae* [9], *Listeria monocytogenes* [10], and *B. cenocepacia* [11]. Besides in silico approaches, experimental procedures include shaving the bacterial cell surface components using proteases, followed by their identification through Mass Spectroscopy (MS) [12]. Another approach named cell attachment blot, used by McClean et al. [13] to identify cell adhesion molecules of *Burkholderia cenocepacia* BC7, uses the immobilization of bacterial membrane proteins after their separation on a 2D electrophoresis gel. They are then incubated with lung epithelial cells followed by their detection with antibodies. Positive spots are subsequently identified, recovered from the gel, and subjected to MALDI-TOF analysis. An adaptation of this same technique, named blot rolling assay, permits the identification of adhesion events between bacteria–host cells under fluid shear conditions. This assay has the advantage of verifying adhesion under flow conditions instead of promoting static interactions [14]. 

Following the identification of adhesins and their experimental validation, it is necessary to use technology to quantify the adhesion forces between bacteria–host cells or host cell components. The use of atomic force microscopy (AFM) proves to be a powerful tool that generates topographical images at the nanometer scale and ensures the precise quantification of the forces exerted. Using AFM cantilevers functionalized with a single bacterium or a purified adhesin, it is possible to scan over a sample surface, whether it consists of host cells, purified host receptors, other adhesins, or extracellular matrix components [15,16].

### 1.1. Adhesion Molecules of B. cenocepacia: State of the Art

*B. cenocepacia* possesses a large repertoire of virulence factors, including adhesins, invasins, enzymes, toxins, siderophores, and secretion systems [7,17]. Focusing on the adhesion process, the number of molecules documented as adhesins is still quite small. The identification of this subset of genes is critical to understand the interactions of bacteria with eukaryotic host cells and may contribute to the future development of novel therapies. To date, the list of known/predicted *B. cenocepacia* J2315 adhesins include: the giant cable pili (cblA); the 22-kDa adhesin (adhA) [18]; the peptidoglycan-associated lipoprotein (BCAL3204) [19]; OmpW-like proteins (BCAL0198, BCAL0287, BCAM0081, and BCAM2346) [13]; and a group of trimeric autotransporter adhesins (BcaA, BcaB, BcaC, BCAM1115, BCAM2418, BCAS0236, and BCAS0335) [20]. Besides this list, there are also described in *B. cenocepacia* proteins called lectins (BclA, BclB, BclC) (carbohydrate-binding proteins), which, due to their ability to bind to sugars, contribute to the attachment/recognition of the bacteria to the host cell and/or components of the extracellular matrix (ECM). These proteins are excreted by the bacterial cells and remain associated with their surfaces, thus promoting adhesion processes. Because they do not represent adhesion factors anchored to the outer membrane and are capable of promoting physical contact between cells, they were not included in the group of adhesins characterized in this study.

To evaluate the involvement of the *cblA* and *adhA* genes in adhesion, Urban et al. [18] constructed isogenic mutants. *cblA* is part of an operon consisting of five genes required for the synthesis of the cable pilus, which is a distinctive feature of the *B. cenocepacia* epidemic lineage ET-12 [21]. The *adhA* gene, located outside the *cbl* operon, encodes a large protein that after a proteolytic event on its N-terminal originates the 22-kDa adhesin, containing hemagglutinin domains. Relative to the parental strain, the *cblA* gene-deficient strain was shown to have a decreased ability to bind cytokeratin CK13 (50%) while the *adhA* mutant lost the binding completely. Although the expression of both genes seems to be independent, the existence of the cbl pili guarantees the maximal binding of *B. cenocepacia* to its CK13 receptor. Moreover, since CK13 is expressed at high levels on the apical surface of airway epithelial cells of CF patients, *B. cenocepacia* strains expressing *cblA* and *adhA* genes may have an increased potential to cause infection [18]. 

Dennehy et al. [19] have demonstrated the direct involvement of the *bcal3204* gene product in the cell adhesion process. *B. cenocepacia bcal3204* encodes a peptidoglycan-associated lipoprotein (Pal) previously studied due to its immunogenicity properties [22]. Subsequently, the construction of a Pal deletion mutant as well as the heterologous expression of the *bcal3204* gene in an *E. coli* host allowed for demonstrating its association with the cell adhesion process [13]. The Pal lipoprotein shares a high similarity with OmpA which has been associated with the events of cellular adhesion in many pathogens [19]. In agreement with these results, O’Grady and Sokol [23] identify *bcal3204* among a list of *B. cenocepacia* genes that were upregulated in an in vivo assay in a rat chronic respiratory infection model. 

Using the cell attachment blot technique, McClean et al. [13] have identified in the *B. cenocepacia* BC7 strain two proteins (OmpW family protein and Linocin-like protein) that show high-affinity adherence to lung cells and are thought to participate in the adhesion process. Blast analysis of the OmpW family protein against the annotated proteins sequences of *B. cenocepacia* J2315 reveals the existence of four hits with a significant identity, namely BCAL0198 (100%); BCAL0287 (36.5%); BCAM0081 (37.5%); and BCAM2346 (61.7%). Validation assays are now needed to confirm the involvement of these proteins as being part of the adhesiome of this pathogenic bacterium. On the other hand, no orthologs of the *B. cenocepacia* BC7 gene encoding for the linocin-like protein were found to exist in the *B. cenocepacia* J2315 genome.

Among the cell adhesion molecules identified in *B. cenocepacia* J2315, the class of trimeric autotransporter adhesins (TAAs) is one of the most studied [24]. Following the release of the genome sequence of B. *cenocepacia* J2315 [25], Mil-Homens et al. [20] identified in the vicinity of the denominated *B. cenocepacia* island (cci) that is a marker of transmissibility [26], a cluster containing three TAAs (*bcam0219*, *bcam0223*, and *bcam0224*; renamed as *bcaA*, *bcaB*, and *bcaC*). In addition to these genes, there are four other TAA genes (*bcam1115*, *bcam2418*, *bcas0236*, and *bcas0335*) located at different chromosomes/positions in the genome of this bacterium. TAAs are outer membrane cell appendages distributed across Gram-negative bacteria and proved to be associated with virulence. Besides their direct involvement in adhesion, TAAs have been demonstrated to represent multifunctional factors, facilitating invasion and immune evasion. The assembly of TAAs use a type Vc secretion system and all contain the same domain architecture, i.e., a C-terminal β-barrel anchor, a passenger that is secreted to the cell surface and an N-terminal signal sequence [27]. 

Although showing functional redundancy, the seven TAA genes differ in how they are regulated. By analyzing their transcription levels after promoting bacterial contact with host cells, Pimenta et al. [11] divided the seven TAA genes into two groups (with higher and lower levels of expression). To uncover the roles of TAAs, a multidisciplinary approach including bacterial genetics, molecular and cell biology, biochemistry, and biophysics was carried out. The cluster of TAAs (*bcaA*, *bcaB*, and *bcaC*) encode proteins with adhesion properties reflected in their abilities to bind to host cells, to components of the extracellular matrix and to form biofilms. In addition, they participate in motility, immune system evasion and in the virulence of the bacterium evaluated in a non-mammalian in vivo system (*Galleria mellonella*) [28,29,30]. It was also possible to use atomic force microscopy to unravel the adhesion mechanisms of the BcaA protein used as a prototype for TAAs. It was shown that BcaA establishes homophilic interactions *in trans*, which are necessary for the auto-aggregation process. It was also found that BcaA binds collagen, a major component of the ECM. Finally, it was possible to demonstrate the binding of BcaA to receptors of live cells (pneumocytes) leading to the formation of membrane tethers necessary for cellular adhesion [30]. In another study, Pimenta et al. [31], using an array of glycans, demonstrated for the first time the binding preferences for the adhesin BCAM2418. BCAM2418 favored binding to 3’sialyl-3-fucosyllactose, histo-blood group A, α-(1,2)-linked Fuc-containing structures, Lewis structures, and GM1 gangliosides. 

### 1.2. Comparative Transcriptomic Analysis to Identify New Candidate Adhesion-Related Genes 

Pimenta et al. [11] have developed an RNASeq-based approach that generated a holistic picture of the adhesion phenomena gene expression profile. The transcriptome of *B. cenocepacia* K56-2 was analyzed after the first contact (30 min) of the bacteria with bronchial host cell-derived vesicles [11]. Genes involved in pathways such as the TCA cycle and glycolysis were downregulated, while genes involved in sulfur and nitrogen metabolism were upregulated. Specifically in relation to adhesion, Pimenta et al. [11] demonstrated the overexpression of some of the genes previously described as adhesion-related in Burkholderia and other Gram-negative bacteria, namely genes encoding components of Flp type pilus, adhA cable pilus associated adhesin, and the BCAM2418 and BCAS0236 trimeric autotransporter adhesins. Furthermore, the study surprisingly revealed the existence of a set of genes highly overexpressed in response to bacterial/host membrane contact that had not been previously described in this bacterium (*bcal1523*, *bcal1524*, and *bcam0695*). They represent genes encoding lipoproteins containing the Collagen Middle domain (InterPro signature—PF15984). Given the particularity of the gene expression pattern verified for this set of genes in response to adhesion, in the second part of this study a computational analysis was carried out to proceed to its characterization.

## 2. Materials and Methods

The proteins of interest were identified by searching for sequences possessing the Bacterial Collagen Middle Region (Collagen_mid) family (PF15984) in one strain representative of each species of the Bcc. Protein sequences were obtained using the Burkholderia Genome Online Database (https://www.burkholderia.com/, accessed on 13 March 2023) [32]. The subcellular localization of each protein was predicted using the PSORTb program (https://www.psort.org/psortb/, acessed on 13 March 2023) [33]. The sequences of amino acids and nucleotides (respectively to collagen-like domain-containing proteins and 16SRNA) were used to obtain phylogenetic trees with phylogeny.fr using the “one-click workflow” (MAFFT for multiple alignments, BMGE for alignment curation, PhyML for tree inference, and Newick Display for tree rendering) [34]. Specific domains and motifs were searched using the Pfam database (pfam.xfam.org, acessed on 13 March 2023) and InterPro online tool (https://www.ebi.ac.uk/interpro/, acessed on 13 March 2023). To identify the tandem repeats, the XSTREAM Web Interface tool was used with default settings [35]. For disordered region prediction, the flDPnn webserver (http://biomine.cs.vcu.edu/servers/flDPnn/, acessed on 13 March 2023) [36] and the SMART webserver (http://smart.embl-heidelberg.de/, acessed on 13 March 2023) [37] was used. The three-dimension (3D) structural prediction of each domain (N-terminal, Collagen-mid, and C-terminal) of the *B. cenocepacia* J2315 proteins were downloaded from the AlphaFold protein structure database (https://alphafold.ebi.ac.uk/, acessed on 13 March 2023) [38,39]. The Yasara program was used to visualize the 3D structures of the proteins. 

## 3. Results and Discussion

### 3.1. Computational Analysis of Collagen-like Domain-Containing Proteins in Members of Bcc Bacteria

#### Distribution, Phylogeny, and Variable-Number of Tandem-Repeat Analysis

Using the Interpro database (https://www.ebi.ac.uk/interpro/, acessed on 13 March 2023), we assessed the presence of the Col_mid_reg (PF15984) across the entire database. The proteins containing this domain are mostly of bacterial origin showing a distribution across the classes of Betaproteobacteria (1690 sequences in 409 species) and Gammaproteobacteria (563 sequences in 380 species). Within the Betaproteobacteria, the Burkholderiaceae family is the most represented (1500 sequences in 291 species). Considering the search in Bcc members, a total of 75 hits were identified. 

First, we analyzed the sequence distribution and the subcellular localization of the 75 Bcc-CLPs (Figure 1). The number of proteins per Bcc species is between 3 and 5. They are predicted to be mostly lipoproteins anchored to the outer or inner membrane.

Then, the 75 Bcc-CLP sequences were used to construct the phylogenetic tree shown in Figure 2. The tree revealed the existence of four clusters (I, II, III, and IV), with cluster IV subdivided into three subclusters. We find that the sequences are not grouped by species, but the division is rather based on the sequence identity of the Col_mid_reg. Proteins from the same cluster possess identities of 80–90%, while proteins from different clusters can have identities of 40%. It is notable that sequences from the same species are distributed through different clusters, which indicates that the Col_mid_reg is variable within the same species.

It is interesting to note that the Bcc-CLPs studied here, although annotated at the InterPro database as containing the so-called “collagen mid-region”, represent a distinct group that exhibits less than 15% pairwise sequence identity with the known well-described collagen-like proteins (CLPs) from bacterial origin. Such bacterial CLPs share, with less or greater extension, structural and functional features with collagens of metazoan origin [40]. Eukaryotic collagen occurs with a triple helical structure revealing a characteristic pattern of continuous GXX’ repeats, in which the first and second X positions are mostly occupied by prolines and hydroxyprolines, respectively. Hydroxyproline represents post-translational modifications of proline by the enzyme prolyl-4-hydroxylase [40]. The existence of such triplet motifs has a pivotal role in defining the helical structure and their binding properties. Unlike eukaryotic collagen, the bacterial CLPs are represented by a heterogeneous group of proteins in which the presence of the motif GXX’ greatly varies in number and in amino acid composition [40,41]. Moreover, only in a few bacterial CLPs was the formation of triple helices confirmed [42]. They have been identified in an increasing number of species and are generally associated with adhesion functions. 

Computational analysis performed with the Bcc-CLPs did not allow their inclusion in the group of CLPs of bacterial origin described so far. In doing so, we considered two possible scenarios: (i) the group of Bcc-CLPs is another example of the heterogeneity found in bacterial collagen-like proteins or (ii) indicate that it is a group of proteins containing CLDs but representing a distinct category. In fact, apart from its annotation in databases, there is no data demonstrating its functionality and similarity with either collagens of eukaryotic origin or the bacterial CLPs of Gram-positive bacteria. They present three distinctive features: (i) their occurrence mostly restricted to members of the Burkholderiaceae family; (ii) the singularity of the existence of the annotated InterPro domain “collagen mid-region”, exhibiting a high frequency of non-consecutive sets of GXX’ triplets; and (iii) the existence of a high number of tandem repeats (TR), mainly at the N-terminal part, that originate regions of low complexity and in which the GTS motif is overrepresented. Based on those singularities, we perform in this study, for the first time, a characterization of these proteins defining them as a group apart from those previously indicated as CLPs of bacterial origin.

Since it has been documented that the existence of TR in proteins promotes their function as adhesion molecules [43,44], next we have analyzed the presence and frequency of TRs in the 75 Bcc-CLPs (XSTREAM web server [35]). As shown in Figure 3, the existence of TR in the majority of the 75 Bcc-CLPs is notable, being absent in only five proteins (6%). Moreover, it deserves a mention that Bcc-CLPs from *B. cenocepacia* J2315 and *B. multivorans* ATCC 17616, the two most problematic Bcc pathogenic species, possess the highest percentages of TR (Figure 3A). We also observed that the frequency of TR is variable between orthologs (Figure 3B). This observation suggests that the TR were differentially accumulated across Bcc species suggesting that they might be correlated with different evolutionary pressures. 

### 3.2. Bcc-CLPs in Burkholderia cenocepacia

Next, we focused our analysis on *B. cenocepacia* J2315 (reference strain; ET-12 lineage) [21], where five Bcc-CLPs were identified: BCAL1523, BCAL1524, BCAL2397, BCAM0695, and BCAM1598. These proteins are all characterized as lipoproteins, anchored either at the inner or outer membrane, indicating their potential as having a role in the adhesion process. 

First, using the Burkholderia.com platform [32], we evaluated the distribution of sequences in the genome and the genes that exist in their vicinity (Figure 4A,B). *B. cenocepacia* J2315 possesses three chromosomes [25], although the five genes of interest are distributed in chromosome 1 (*bcal1523*, *bcal1524*, and *bcal2397*) and chromosome 2 (*bcam0695* and *bcam1598*) (Figure 4A).

*bcal1523* and *bcal1524*, together with the neighbors *bcal1522* and *bcal1520*, may be part of an operon. These four genes are flanked upstream by a transposase (*bcal1519*) and downstream by the *flp* pilus operon, being a virulence trait described in other organisms [45,46]. The *bcal2397* is upstream flanked by a hypothetical protein (*bcal2396*) and by the cafA (Ribonuclease G) encoding gene (*bcal2395*). The downstream region of *bcal2397* contains a lipoprotein with an unknown function (*bcal2398*) followed by a major facilitator superfamily protein (*bcal2399*) and a Fusaric acid resistance protein-like (*bcal2400*). Mapped in the upstream vicinity of *bcam0695*, a gene is located (*bcam0694*) encoding an integral protein membrane sharing primary sequence identity at its C-terminal region with the Isoprenylcysteine carboxyl methyltransferase (ICMT) family of enzymes [47]. It may be possible that this ortholog of ICMT acts on the Bcc-CLP (BCAM0695) by promoting its lipidation to the membrane environment. The downstream region of *bcam0695* contains a gene encoding a carboxymuconolactone decarboxylase probably involved in the degradation of aromatic compounds. The *bcam1598* gene neighborhoods contain a downstream gene coding for an AsnC family regulatory protein and upstream a gene encoding for a protein involved in copper and silver resistance (Figure 4B and Table 1). 

Using InterPro (https://www.ebi.ac.uk/interpro/, accessed on 13 March 2023) and the SMART platforms [37], we have characterized the domain architecture of each of the five Bcc-CLPs from *B. cenocepacia* J2315. Each of them contains a bacterial collagen mid-region domain with a size of approximately 200 amino acids. In addition, low complexity regions (LCR) are surprisingly represented in high numbers and distributed preferentially in the vicinity (N and C terminus) of the collagen mid-region domain (Figure 5). The percentage of these LCR is variable, with a maximum of 72% for the BCAM0695 protein and a minimum of 43% for the BCAL1523 (Table 2). 

Analyzing the LCR, we observed that the most frequent combination of amino acids is GTS, comprising 91% of the GXX’ tripeptides present in these proteins. We found that this motif is present in four out of five CLPs, with BCAL2397 as the exception. The number of GXX’ tandem repeats vary between 4 in BCAL1523 and 83 in BCAM0695 and are preferentially located at the N terminal part of the proteins (Table 2). To date, the functional significance of these extended sets of GTS tandem repeats remains unclear. However, either in eukaryotes or prokaryotes, a link exists between the proteins containing tandem repeats and their direct involvement in cell adhesion [43,48]. In addition, it has been described that surface-exposed serine-rich repeats glycoproteins from Gram-positive bacteria promote adherence to multiple host cell receptors [49]. Overall, we believe that the existence of GTS tandem repeats in Bcc-CLPs follows a sequence-based evolutionary framework that somehow contributes to the adhesive role of these proteins.

Finally, we used the program flDPnn [36] to investigate the possible correlation between the existence of many TRs and the intrinsically disordered prediction levels of these proteins. Intrinsically Disordered Proteins (IDPs) are defined as lacking stable secondary and/or tertiary structures and are enriched with TR [50]. As shown in Figure 6, all five Bcc-CLPs exhibit high levels of disorder, which coincide with the locations of the TR. Furthermore, the predicted disordered regions possess a high score for protein, DNA or RNA binding, indicating their putative importance in binding events [51]. In contrast, the ordered regions of the Bcc-CLPs coincide with the existence of their collagen mid-region domains. Using the AlphaFold protein structure database [38,39], we obtained a model for each protein truncate, which allowed the visualization of the three parts of each Bcc-CLP (N-terminal, Collagen Middle Region, and C-terminal) (Figure 6).

Initially, IDPs were thought of as being uncommon and unspecific, but they are now identified in all kingdoms, being involved in several biological functions [52]. In the case of bacteria, it is estimated that the percentage of IDPs or proteins with disordered regions varies from 18 to 28%, a relatively small amount compared with eukaryotic organisms [53]. In terms of function, for bacteria, Peng et al. [51] identified a correlation between intrinsically disordered regions and DNA and RNA binding, as well as sporulation, catabolic and metabolic processes, and pathogenesis. Mier et al. [54] also suggest that LCR are more represented in the outer membrane and extracellular proteins in bacteria, being involved in host–pathogen interactions. These discoveries unravel the importance of disorder and LCR in proteins associated with pathogenicity and adhesion processes, encouraging further study of the importance of these domains in CLPs.

## 4. Conclusions

This study aims to highlight the biological relevance of the adhesion events as a first step in respiratory infections caused by the bacterium *B. cenocepacia*. To this end, in the first part, an updated review of the state-of-the-art regarding the adhesion factors that are known in this pathogenic bacterium is presented. It is highlighted that the adhesion process is multifactorial, synergistic, and considered critical for the success of the infection. In *B. cenocepacia*, given its intrinsic nature for antibiotic resistance, knowledge of which and how adhesion factors function is of utmost relevance to finding the complementary forms of therapies. Adhesion factors are membrane proteins, exposed on the outside of the bacterium and able to establish more or fewer specific bonds either with components of the extracellular matrix or with host cell receptors.

In an attempt to find new adhesion factors used by the pathogenic bacterium *B. cenocepacia* J2315, the recent work published by Pimenta et al. [11] highlights the existence of a set of not yet studied lipoproteins, containing collagen-like domains and predicted by PSORT to be anchored, respectively, to the outer membrane (BCAL1524, BCAL2397, BCAM0695, and BCAM1598) or cytoplasmic membrane (BCAM1523). The existence of the collagen domains plus their predicted cellular localization suggests their participation in the adhesion process. In this work, for the first time, we proposed to perform a comprehensive computational analysis of this group of proteins considered relevant for adhesion and most likely representing an addition to the adhesin portfolio of *B. cenocepacia*. We surprisingly found that although annotated as CLPs, they are not homologous to proteins with the same annotation as the ones that exist in bacteria. The known bacterial CLPs are more or less identical to collagens of animal origin, both in their amino acid sequence (presence of GXX’ repeats) and in their three-dimensional structure, characterized by the formation of triple helices. 

Based on our study, we can state that the Bcc-CLPs are almost exclusively distributed in members of the Burkholderiaciae family. They contain a domain annotated as “Collagen mid-region” (InterPro signature—PF15984) and downstream and upstream a very significant set of TR. Among these repeats, the tripeptide GTS is overrepresented corresponding to a high percentage of their primary sequences. Thus, based on our computational study, we are not able to conclude about the nature of this particular group of proteins. Based on the significant variability in amino acid sequences and the structural features found in CLPs of bacterial origin, Bcc-CLPs may be a new member to be included, or alternatively represent a distinct group of proteins that although containing collagen-like domains, do not belong to the collagen family.

Finally, it is worth mentioning their singularity regarding the presence of numerous TRs of the GTS tripeptide, which attributes to these proteins’ extensive regions of structural disorder. It is thus pertinent to ask what the relevance of TRs is and how they contribute to the putative adhesive functions of these proteins. We speculated that the existence of intrinsically disordered regions in CLPs gives them conformational plasticity and thus favors their adhesive properties.

The results of the computational analysis described here now justify carrying out a set of experimental assays, namely the construction of mutants and their phenotypic evaluation as well as the heterologous expression of each gene followed by the purification of CLPs to clarify their functions in the cell adhesion process.

## Figures and Tables

**Figure 1 microorganisms-11-01118-f001:**
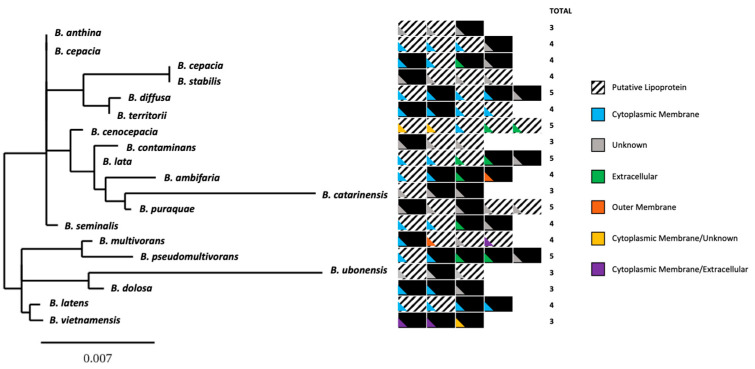
Distribution of the 75 Bcc-CLPs per species and with the indication of their putative subcellular localization. The phylogenetic tree was constructed using the 16S RNA gene sequences. The color code indicates the putative localization of each protein as predicted by P-SORTb [33]. Squares black and white represent lipoproteins and black squares are non-lipoproteins.

**Figure 2 microorganisms-11-01118-f002:**
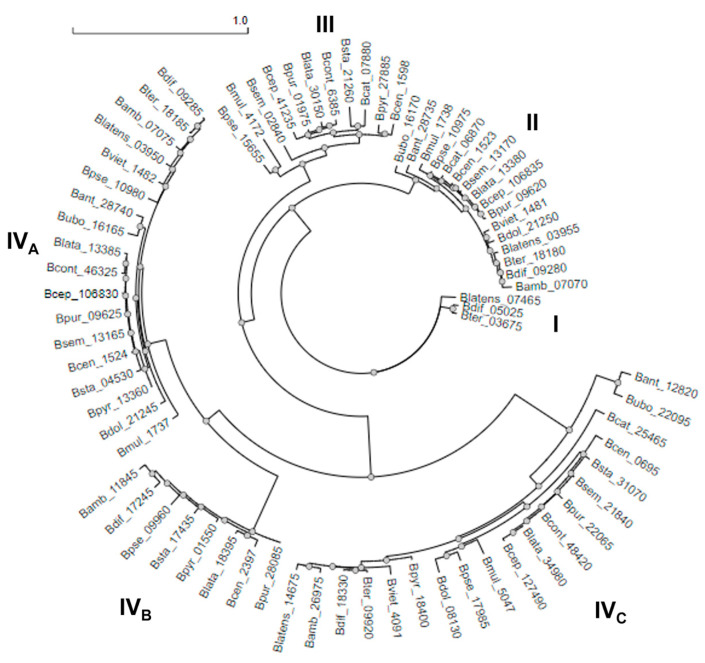
Phylogenetic tree of the 75 putative Bcc-CLPs, divided into four clusters: I, II, III and IV (with three sub clusters). The division is based on the identity of the Col_mid_reg.

**Figure 3 microorganisms-11-01118-f003:**
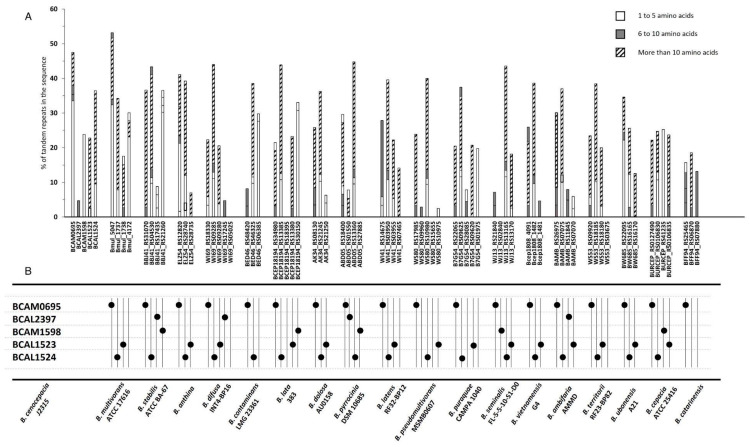
(**A**)—Distribution and frequency of tandem repeats in the 75 Bcc-CLPs. (**B**)—Analysis of orthologues of Bcc-CLPs from *B. cenocepacia* J2315 across the other Bcc species. Filled circles represent the presence of the orthologue.

**Figure 4 microorganisms-11-01118-f004:**
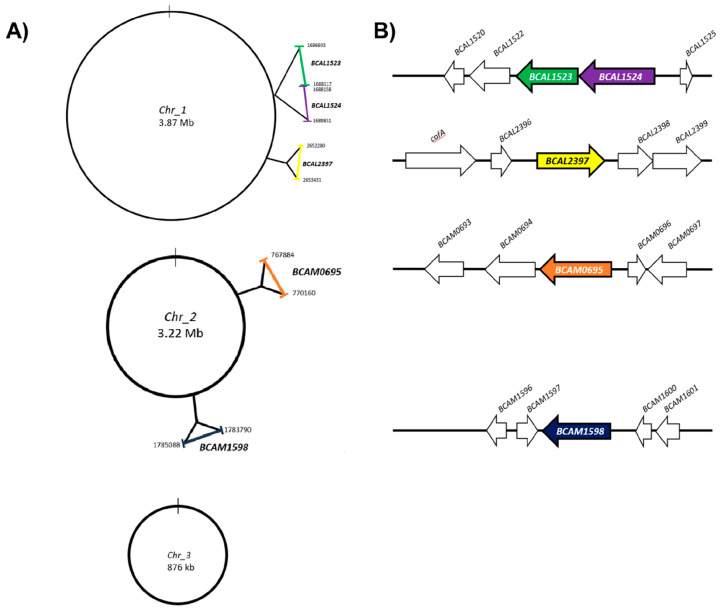
(**A**) Distribution and localization of the five collagen-like encoding genes in the *B. cenocepacia* J2315 chromosomes. (**B**) Vicinity of the five Bcc-collagen-like genes (see also Table 1).

**Figure 5 microorganisms-11-01118-f005:**
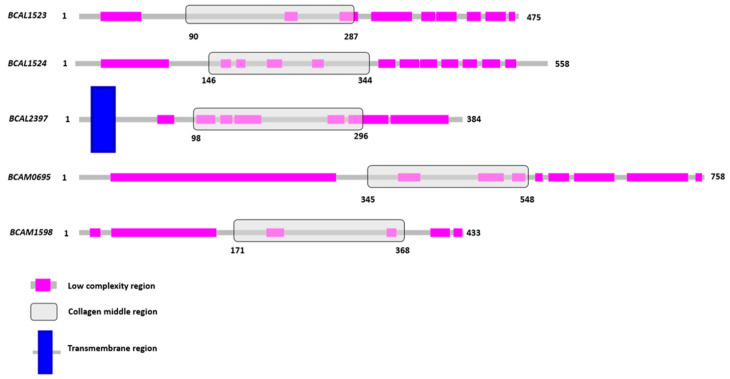
Representation of the low complexity regions and the bacterial collagen middle region predicted by SMART and InterPro webservers.

**Figure 6 microorganisms-11-01118-f006:**
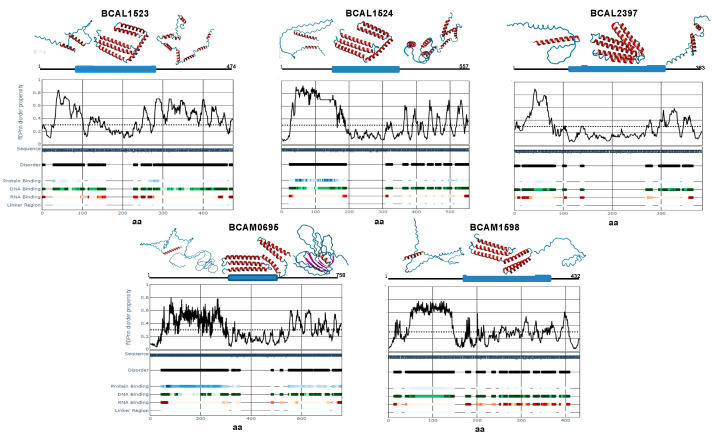
Disorder scores obtained with the flDPnn webserver and AlphaFold structure predictions.

**Table 1 microorganisms-11-01118-t001:** Genomic localization of Bcc-CLP encoding genes as well as genes/products located in their vicinity. + and - represent the two strands of DNA.

Gene Name	Genomic Location	Strand	Cellular Localization	Product Description
*bcal1519*	1683715–1684125	+	Periplasmic	Putative Transposase
*bcal1520*	1684227–1684688	-	Periplasmic	Putative Lipoprotein
*bcal1522*	1684863–1686548	-	Outer Membrane	Putative Exported Heme Utilization-Related Protein
** *bcal1523* **	**1686693–1688117**	**-**	**Cytoplasmic Membrane**	**Bcc-CLP**
** *bcal1524* **	**1688158–1689831**	**-**	**Extracellular**	**Bcc-CLP**
*bcal1525*	1690509–1690679	+	Unknown	Flp type Pilus Subunit
*cafA*	2649893–2651362	+	Cytoplasmic	Ribonuclease G
*bcal2396*	2651489–2651824	+	Unknown	Hypothetical Protein
** *bcal2397* **	**2652280–2653431**	**+**	**Extracellular**	**Bcc-CLP**
*bcal2398*	2653685–2654650	+	Unknown	Putative Lipoprotein
*bcal2399*	2654647–2655906	+	Cytoplasmic Membrane	Major Facilitator Superfamily Protein
*bcal2400*	2655945–2656511	+	Cytoplasmic Membrane	Fusaric acid resistance protein-like
*bcam0693*	765308–765925	-	Cytoplasmic	Hypothetical Protein
*bcam0694*	766473–767756	-	Cytoplasmic Membrane	Hypothetical Protein
** *bcam0695* **	**767884–770160**	**-**	**Unknown/Cytoplasmic Membrane**	**Bcc-CLP**
*bcam0696*	770523–770921	+	Unknown	Putative Carboxymuconolactone Decarboxylase
*bcam1597*	1783216–1783677	+	Cytoplasmic	AsnC Family Regulatory Family
** *bcam1598* **	**1783790–1785088**	**-**	**Extracellular/Cytoplasmic Membrane**	**Bcc-CLP**
*bcam1600*	1785332–1785679	-	Periplasmic	Copper binding periplasmic protein

**Table 2 microorganisms-11-01118-t002:** Characterization of the five Bcc-CLPs identified in *B. cenocepacia* J2315. The number of amino acids, the predicted localization of the protein, the significance of the existence of the Collagen Middle Region, the percentage of low complexity region per protein, and the number and type of GXX’ repeats.

Protein	Nº aa	Localization	Collagen Middle Region E-Value	Low Complexity Region (%)	Nº GXX’ Repeats	GXX’ Type
BCAL1523	475	Cytoplasmic Membrane	1.8 × 10^−63^	43	4	GTS_3_GSS_1_
BCAL1524	558	Extracellular	5.1 × 10^−68^	49	18	GSS_1_GTS_16_GVS_1_
BCAL2397	384	Extracellular	9.4 × 10^−58^	63	-	-
BCAM0695	758	Unknown/ Cytoplasmic Membrane	9.3 × 10^−56^	72	83	GSS_3_GTS_79_GTG_1_
BCAM1598	433	Extracellular/Cytoplasmic Membrane	1.2 × 10^−56^	45	36	GTN_1_GTG_1_GTS_31_GTP_1_GTN_1_GII_1_

## Data Availability

Not Applicable.

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
