# Peer review of "In-Silico Analysis Highlights the Existence in Members of Burkholderia cepacia Complex of a New Class of Adhesins Possessing Collagen-like Domains"

_microorganisms, 2023, doi:10.3390/microorganisms11051118_

Round 1

Reviewer 1 Report

The work submitted by Estevens et al., aims to describe via an in silico approach a new class of adhesins from Burkholderia spp. belonging to the Bcc. This work is interesting and gives new insight into an understudied type of adhesins. I suggest the following clarifications before acceptance:

In lines 358-361: the authors are stating that "in this study" they demonstrated in adhesion assays the adhesion of type II CLPs. Due to the increased expression of the adhesin (not sure if it's about the gene or the protein). I did not find any results about this claim in the paper nor any information about the method in the materials and methods section. My guess is the authors are referring to reference 26 (involving one author of the present work) as most part of the conclusion. Please clarify this point to help the readers to understand what makes this current work new. 

Likewise, transcriptomic analyses are giving clues about the expression of the corresponding proteins but nothing about the functions. Please rephrase accordingly.

Minor:

Please check the consistency of the paper. For example, gram is sometimes in uppercase (lines 346-347) and sometimes not (line 32). It should be without.

Author Response

We appreciate the comments/criticisms to our submitted manuscript.

We have submitted a manuscript that should satisfy the comments/criticisms that were raised by the Referees; specific responses outlining the changes are listed below. We believe that the changes/corrections made have improved our manuscript, and we thank the referees for all the suggestions.

We hope that all the revisions and changes are in your accordance, and we believe that our work now meets the standards of Microorganisms. We appreciate the time and effort that you devoted to the revision of our work.

Yours sincerely,

Arsénio M Fialho

Answers to the reviewer’s comments

Each of the reviewer’s remarks is mentioned, followed by our comments:

Reviewer #1

Major Comments:

The work submitted by Estevens et al., aims to describe via an in silico approach a new class of adhesins from Burkholderia spp. belonging to the Bcc. This work is interesting and gives new insight into an understudied type of adhesins. I suggest the following clarifications before acceptance:

We thank the Referee for his/her positive appreciation of our work and useful comments.

In lines 358-361: the authors are stating that "in this study" they demonstrated in adhesion assays the adhesion of type II CLPs. Due to the increased expression of the adhesin (not sure if it's about the gene or the protein). I did not find any results about this claim in the paper nor any information about the method in the materials and methods section. My guess is the authors are referring to reference 26 (involving one author of the present work) as most part of the conclusion. Please clarify this point to help the readers to understand what makes this current work new. 

Likewise, transcriptomic analyses are giving clues about the expression of the corresponding proteins but nothing about the functions. Please rephrase accordingly.

We agree with the Reviewer comments and in this revised version we rewritten the text (section “Comparative transcriptomic analysis to identify new candidate adhesion-related genes”, page 4 and 5) to highlight the relevance of Pimenta et al (reference 26) findings and to clarify the motivation of this study.

Minor:

Please check the consistency of the paper. For example, gram is sometimes in uppercase (lines 346-347) and sometimes not (line 32). It should be without.

Corrected as indicated

Reviewer 2 Report

Estevens et al. present a manuscript detailing a bioinformatic analysis into putative collagen-like proteins of the Burkholderia cenocepacia complex. These proteins were previously identified through transcriptomics as being upregulated during adhesion to host cells. In addition, the authors give an overview of known adhesion mechanisms of B. cenocepacia

While I do not doubt that the bioinformatics has been conducted properly, I find this to be a rather unsatisfactory manuscript that would need to undergo major revisions before it could be considered for publication. I'm not sure the rather cursory review of adhesion mechanisms adds much to the paper - this could be written as a separate, and more detailed, review, and focus on the CLPs here. 

Regarding the CLPs themselves, very little background is given on CLPs in bacteria, including why they are considered collagen-like. I suspect this is because of an abundance of G-X-X' triplets, which (when containing a high proportion of imino acids) can lead to the formation of a collagen-like polypropylene II triple-helical structure. Note that G-X-X' is preferable to the notation the authors use (GXY), as the Y can easily be confused with tyrosine. None of this background is explained in the text.

The authors suggest that the Bcc proteins, which contain tandemly repeated GTS motifs, should be considered a new class of CLPs. I disagree, mostly because I do not think these proteins are in any way collagen-like. The lack of imino acids suggests that these proteins are not collagenous. Rather, the 'collagen middle region' present in these proteins appeas largely unstructured, and certainly does not represent a clear domain (as demonstrated by the authors' modelling efforts, where the predicted structures are wildly different). Is there any evidence for the formation of collagen-like triple helices? If not, a reclassification sounds appropriate.

The authors imply (but do not state outright) that these proteins mediate adhesion. They are annotated as lipoproteins. My question here is whether there is any evidence for surface exposure of lipoproteins in B. cenocepacia? The a priori assumption here would be that these are periplasmic proteins that may be upregulated as a stress response or similar. This should be addressed in the discussion.

In addition, I have the following minor comments:

1. Moderate English revision is needed. For example, line 33: "... closely related genetically and phenotypically bacteria" is clumsy and misleading (bacteria are not related phenotypically, but they can have similar phenotypes). Similar examples are found throughout the text, please check carefully. Please also capitalise Gram in Gram-negative (it is a name, after all).

2. Line 119: 'high homology' - this is incorrect, as homology is not a quantitative trait; two genes/proteins either are homologous (derived from a common ancestor), or they are not. 'High similarity' or 'high identity' would be correct here.

3. Line 137: please explain 'cci island'. There also seems to be something missing after 'transmissible'.

4. line 142: a reference could be added for TAAs.

5. line 174: the Flp pilus is not covered in the preceding review - why is that?

6. line 200: I-TASSER was used for modelling. I wonder why Alphafold2 was not employed here, as this now represents the state of the art.

7. line 207: 'in 57 of 85 identified species'. This is rather confusing - are the authors suggesting there are only 85 species in the Betaproteobacteria?

8. Figure 1: This could be improved, not least by writing scientific names in italics. The legend should explain how the tree was constructed (16S?) and what black boxes mean on the right.

9. To demonstrate differences to known CLPs, a sequence alignment of some representative proteins should be included.

10. Figure 3: In addition to the information presented, it would be helpful if the analysis were taken further to show which of these proteins represent orthologous groups (i.e. proteins from differents species/strains but found in the same genetic locus). Are there similarities between orthologues? This would also be helpful in coming up with a naming scheme for the proteins.

11. line 249: how is it known these are surface-exposed lipoproteins?

12. The functions mentioned in the section between lines 259-283 could be indicated in figure 4 as well.

13. line 295: as far as I can tell, it is speculation that these proteins are (directly) involved in adhesion. Please modify accordingly.

Author Response

We appreciate the comments/criticisms to our submitted manuscript.

We have submitted a manuscript that should satisfy the comments/criticisms that were raised by the Referees; specific responses outlining the changes are listed below. We believe that the changes/corrections made have improved our manuscript, and we thank the referees for all the suggestions.

We hope that all the revisions and changes are in your accordance, and we believe that our work now meets the standards of Microorganisms. We appreciate the time and effort that you devoted to the revision of our work.

Yours sincerely,

Arsénio M Fialho

Answers to the reviewer’s comments

Each of the reviewer’s remarks is mentioned, followed by our comments:

Reviewer #2

Regarding the CLPs themselves, very little background is given on CLPs in bacteria, including why they are considered collagen-like. I suspect this is because of an abundance of G-X-X' triplets, which (when containing a high proportion of imino acids) can lead to the formation of a collagen-like polypropylene II triple-helical structure. Note that G-X-X' is preferable to the notation the authors use (GXY), as the Y can easily be confused with tyrosine. None of this background is explained in the text.

We understand and agree with the Reviewer, and in this revised manuscript the text has been changed to include information about the collagens from eukaryotic and prokaryotic origin as well a comparison between them. The text has been rewritten to make its meaning clear:

- Page 7 (1st and 2nd paragraphs)

The authors suggest that the Bcc proteins, which contain tandemly repeated GTS motifs, should be considered a new class of CLPs. I disagree, mostly because I do not think these proteins are in any way collagen-like. The lack of imino acids suggests that these proteins are not collagenous. Rather, the 'collagen middle region' present in these proteins appeas largely unstructured, and certainly does not represent a clear domain (as demonstrated by the authors' modelling efforts, where the predicted structures are wildly different). Is there any evidence for the formation of collagen-like triple helices? If not, a reclassification sounds appropriate.

We agree with the Reviewer, and in this revised version we have rewritten the text to reflect that the computational study here performed opens two possibilities: either this group of proteins represents an addition to the bacterial collagen-like proteins or on the contrary it may represent a group of proteins different from collagen. The collagen-like proteins found in bacteria are very diverse in their sequences, GXX motifs (number and sequences, some of which do not contain proline). The formation of the triple helix structure has only been demonstrated for a very small number of proteins (Bachert et al.PLOS ONE | DOI:10.1371/journal.pone.0137578)

- Page 7 (Results and discussion)

- Page 13 (Conclusions)

The authors imply (but do not state outright) that these proteins mediate adhesion. They are annotated as lipoproteins. My question here is whether there is any evidence for surface exposure of lipoproteins in B. cenocepacia? The a priori assumption here would be that these are periplasmic proteins that may be upregulated as a stress response or similar. This should be addressed in the discussion.

In the conclusion section the predicted localization of the Bcc-CLP proteins is discussed in the context of their putative association with adhesion.

Minor comments:

All minor comments were incorporated in this revised version. We thank the Referee for the careful analysis of the manuscript.

  1. Moderate English revision is needed. For example, line 33: "... closely related genetically and phenotypically bacteria" is clumsy and misleading (bacteria are not related phenotypically, but they can have similar phenotypes). Similar examples are found throughout the text, please check carefully. Please also capitalise Gram in Gram-negative (it is a name, after all).

Modify as suggested

  1. Line 119: 'high homology' - this is incorrect, as homology is not a quantitative trait; two genes/proteins either are homologous (derived from a common ancestor), or they are not. 'High similarity' or 'high identity' would be correct here.

Ok

  1. Line 137: please explain 'cci island'. There also seems to be something missing after 'transmissible'.

Ok

  1. line 142: a reference could be added for TAAs.

Ok

  1. line 174: the Flp pilus is not covered in the preceding review - why is that?

In the first part of this manuscript, we describe the state of the art of genes/proteins directly involved in adhesion and described in the species B. cenocepacia. Although the involvement of the Flp pilus has been described in several pathogenic Gram-negative bacteria, in B. cenocepacia there is not yet a published study that demonstrates and characterizes its participation in adhesion. We rewritten the text to clarify this point.

  1. line 200: I-TASSER was used for modelling. I wonder why Alphafold2 was not employed here, as this now represents the state of the art.

We agree with the Reviewer and in light of his/her comment we have now prepared a new Figure 6 using the prediction made by AlphaFold instead I-Tasser. The results obtained showing for all the five proteins an ordered structure for the designated "Collagen-mid region"

  1. line 207: 'in 57 of 85 identified species'. This is rather confusing - are the authors suggesting there are only 85 species in the Betaproteobacteria?

We completely agree with the reviewer. We are very sorry for the confusion. We rewritten the text to correct this description.

  1. Figure 1: This could be improved, not least by writing scientific names in italics. The legend should explain how the tree was constructed (16S?) and what black boxes mean on the right.

We thank the Reviewer, and the figure 1 and legend were modified as suggested.

  1. To demonstrate differences to known CLPs, a sequence alignment of some representative proteins should be included.

In this revised version of the manuscript, we have introduced the information that the amino acid identity between the proteins under study and those described as collagen-like (e.g. Streptococcal Scl1 and Scl2) is of the order of 15%.

  1. Figure 3: In addition to the information presented, it would be helpful if the analysis were taken further to show which of these proteins represent orthologous groups (i.e. proteins from differents species/strains but found in the same genetic locus). Are there similarities between orthologues? This would also be helpful in coming up with a naming scheme for the proteins.

We thank the Reviewer, and the figure 3 was modified to include information related with the orthologous. This information is also included in the text.

  1. line 249: how is it known these are surface-exposed lipoproteins?

We are very sorry for the confusion. We have made some adjustments in the text to clarify the information related with the use of PSort used for the cellular predicition of the lipoproteins.

  1. The functions mentioned in the section between lines 259-283 could be indicated in figure 4 as well.

As suggested this information is now also incorporated in a new Table.

  1. line 295: as far as I can tell, it is speculation that these proteins are (directly) involved in adhesion. Please modify accordingly.

We are very sorry for the confusion. We have made some adjustments in the text to clarify the information.

Round 2

Reviewer 2 Report

The authors have addressed my main concerns. There are still a few points that could be addressed, but as these are mostly typlogical, this can be done in proof.

1. Please write the scientific name in the title in italics.

2. Figure 1. It is still unclear what black squares are. Non-lipoproteins?  Please clarify.

3. Background on CLPs: "... defining the helical structure..."